# Proanthocyanidins Modulate Rumen Enzyme Activities and Protein Utilization In Vitro

**DOI:** 10.3390/molecules27185870

**Published:** 2022-09-10

**Authors:** Sultan Singh, Pushpendra Koli, Brijesh K. Bhadoria, Manjree Agarwal, Suman Lata, Yonglin Ren, Xin Du

**Affiliations:** 1ICAR-Indian Grassland and Fodder Research Institute, Jhansi 284003, India; 2College of Science, Health, Engineering and Education, Murdoch University, 90 South Street, Murdoch, WA 6150, Australia; 3Scientific Service Division, ChemCentre, Cnr Manning Road and Townsing Drive, Bentley, WA 6102, Australia

**Keywords:** flavonoids, proanthocyanidins, *Anogeissus pendula*, *Eugenia jambolana*, ruminal enzymes, rubisco

## Abstract

This study investigated the principal leaf protein (rubisco) solubilization and in vitro ruminal enzyme activity in relation to the molecular structure of proanthocyanidins extracted from leaves of *Anogeissus pendula* and *Eugenia jambolana*. Six proanthocyanidin fractions were extracted by 50% (*v*/*v*) methanol–water followed by 70% (*v*/*v*) acetone–water and then distilled water from leaves of *A. pendula* (AP) and *E. jambolana* (EJ) to yield EJ–70, EJ–50, EJ–DW, AP–70, AP–50 and AP–DW. Fractions were examined for their molecular structure and their effects on sheep ruminal enzymes and solubilization of rubisco in vitro. All fractions significantly (*p* < 0.05) inhibited the activity of ruminal glutamic oxaloacetic transaminase and glutamic pyruvic transaminase. The fractions AP–50 and EJ–50 significantly inhibited the activity of the *R*-cellulase enzyme. Most of the fractions inhibited *R*-glutamate dehydrogenase activity (*p* < 0.05) by increasing its concentration, while protease activity decreased by up to 58% with increasing incubation time and concentration. The solubilization of rubisco was observed to be comparatively higher in *A. pendula* (16.60 ± 1.97%) and *E. jambolana* (15.03 ± 1.06%) than that of wheat straw (8.95 ± 0.95%) and berseem hay (3.04 ± 0.08%). A significant (*p* < 0.05) increase in protein solubilization was observed when wheat straw and berseem hay were supplemented with *A. pendula* and *E. jambolana* leaves at different proportions. The efficiency of microbial protein was significantly (*p* < 0.05) greater with the supplementation of leaves of *A. pendula* in comparison to *E. jambolana*. The overall conclusion is that the proanthocyanidins obtained from *E. jambolana* exhibited greater inhibitory activities on rumen enzymes, whereas *A. pendula* recorded higher protein solubilization. Thus, PAs from *A. pendula* and *E. jambolana* appear to have the potential to manipulate rumen enzyme activities for efficient utilization of protein and fiber in ruminants.

## 1. Introduction

*Anogeissus pendula* Edgew. and *Eugenia jambolana* Lam. are gregarious tree species indigenous to dry and mixed forests of India [1]. These are socially and economically important trees because of their provision of fuel, timber, medicine, and animal fodder, amongst other attributes [2]. Provision of this leaf fodder is valuable in local farming systems, especially during lean periods as sources of protein, energy, oil, fatty acid and mineral supplements for livestock [3,4]. Tree foliage often contains the phenolic compounds polyphenols, tannins, alkaloids, and proanthocyanidins, some of which limit nutrient utilization in ruminants [5,6,7,8], while others increase nutrient utilization [9,10]. The leaves of *Anogeissus pendula* (AP) and *Eugenia jambolana* (EJ) were assessed for crude protein (6.8–10.6%), neutral detergent fiber (46.4–58.6%), lignin (13.5–23.6%), and *in vitro* dry matter digestibility (25–38%) in our previous study [11]. Proanthocyanidins (PAs) from tree leaves are complex bioactive compounds derived from polymerization of flavon-3-ol units. These molecules are complex in nature, which complicates their characterization [12,13]. PAs are considered to have both adverse and beneficial effects, depending on concentration and type, and their interaction with animal factors including the animal species, its physiological state and the wider composition of its diet. 

Extraction of PAs is commonly performed by using appropriate organic solvents in combination with column chromatography, paper chromatography, thin layer chromatography (TLC) and high-performance liquid chromatography (HPLC) (Hammerstone). Characterization of PAs remains problematic due to their diverse chemical nature, higher molecular weight and poor ionization. Thus, UV remains a prominent technique that has been used for many years [14,15]. TLC and paper chromatography remains a versatile method for the detection and separation of phenolics from crude extracts. The spray reagents for the detection of phenolics have been reviewed [16,17,18]. High performance thin layer chromatography (HPTLC) provides an optimal method for separation of complex mixtures [19]. 

Among plant secondary metabolites, phenolics have drawn considerable attention due to the range of their biological activities, notably their antimicrobial properties [20]. Globally, lot of research has been done on effects of phenolic compounds on rumen fermentation and microbiota [21,22]. However, information is scanty on phytochemical properties of phenolics (flavonoids or PAs) in relation to their effects on rumen enzyme activities and rumen fermentation process [23,24,25,26].

In view of these research gaps, in the present study we aimed to extract and characterize PAs from leaves of *A. pendula* and *E. jambolana* to investigate the effects of their molecular structure on in vitro ruminal enzyme activities and solubilization of protein. 

## 2. Results

### 2.1. Extraction and Molecular Characterization of Proanthocyanidins 

The results of our studies on proanthocyanidins, viz., (+) catechin, (−) epicatechin, (+) gallocatechin and (−) epigallocatechin, along with their -4-phloroglucinol adducts in fractions AP–50 (eluted by 50% (*v*/*v*) methanol–water), AP–70 (eluted by 70% (*v*/*v*) acetone–water), AP–DW (eluted by water), EJ–50 (eluted by 50% (*v*/*v)*, EJ–70 (eluted by 70% (*v*/*v*) acetone–water) and EJ–DW (eluted by water), are presented in Table 1. The details of their molecular structures are reported in Appendix A. The catechin and its -4-phloroglucinol adduct were in significantly greater (*p* < 0.05) quantity compared to other Pas, except in AP–DW. Fraction AP–70 was rich (*p* < 0.05) in all components of PAs in comparison to other fractions, whereas (−) epigallocatechin and (+) catechin-4-phloroglucinol were not detected in AP–DW. All fractions varied with the polymeric structure and degree of polymerization. The degree of polymerization varied from 2.66–7.80. The range of extender unit percentage and terminal unit percentage varied from 73–89 and 11–27, respectively. The subunits of anthocyanin in most of the fractions of *E. jambolana* and *A. pendula* were found to be predominantly delphinidin and cyanidin in a ratio of 60:40, whereas additional subunit pelargonidin and pelargonidin with malvidin were found in EJ–DW and AP–50, respectively. Further information is available in Appendix A.

### 2.2. Effect on Rumen Enzymes Activities

Ruminal glutamic oxaloacetic transaminase (*R*–GOT(P), glutamic pyruvic transaminase (*R*–GPT(P) and *R*–GPT(B)) and cellulase activities significantly (*p* < 0.05) decreased in all fractions of PAs (Table 2). Moreover, the EC_50_ of *R*–GOT(B) was significantly higher (*p* < 0.05) in fraction EJ–50 relative to the control, whereas tannic acid showed higher (*p* < 0.05) cellulase activity over the control. Among all fractions, *R*–GOT activity significantly decreased (*p* < 0.05) in EJ–DW and AP–50 in protozoal and bacterial fractions, respectively. In the case of *R*–GPT, the lowest (*p* < 0.05) values were observed in EJ–DW (7.83 ± 1.0) and EJ–50 (3.23 ± 0.26) for protozoal and bacterial fractions, respectively. Fraction AP–50 was more effective (*p* < 0.05) in inhibiting *R*-cellulase in comparison to other fractions.

Ruminal glutamate dehydrogenase (*R*–GDH) activity significantly (*p* < 0.05) decreased with increasing concentration of each PA fraction in both protozoal and bacterial parts compared to the control (Figure 1). Interestingly, the AP–DW fraction exhibited complete inhibition of *R*–GDH activity even at low concentration in both bacterial and protozoal parts. At a concentration of 2 mg/mL of AP–50, activity was significantly higher (*p* < 0.05) than the control in protozoal and bacterial parts. Fraction AP–50 reduced *R*–GDH protozoal activity to the extent of 5.69 ± 0.87 IU/L at 14 mg/mL from 35.69 ± 1.55 IU/L at 2 mg/mL. A similar trend was observed in all fractions with increasing concentration in both bacterial and protozoal parts. Fraction EJ–DW was able to reduce the bacterial *R*–GDH from 28.61 ± 0.85 IU/L at 2 mg/mL to 13.82 ± 0.74 IU/L at 14 mg/mL. Fraction EJ–70 had more potential to inhibit *R*–GDH bacterial fraction activity, and at 10 mg/L it completely halted *R*–GDH activity.

*R*–protease activity significantly (*p* < 0.05) decreased (Figure 2) in all fractions with each increasing concentration and incubation time in comparison to the control. Fraction AP–50 at different concentrations and periods of incubation reduced protease activity in the ranges of 23–52% and 7–33%, respectively. In the case of AP–70, the decline in protease activity was observed between 10–51% and 3–43% with each incubation time and concentration. Fractions EJ–50 and EJ–70 recorded 9–58% and 7–39% reduction at each concentration, respectively. Based on concentration and incubation period, fraction EJ–50 was the most effective in the reduction of proteolytic activity (0.11 ± 1.09 µg/min/mL) followed by AP–50, EJ–70 and AP–70.

### 2.3. Effect on Protein Solubilization and Microbial Protein Efficiency

Dietary supplementation of *E. jambolana* and *A. pendula* leaves in different ratios with wheat straw and berseem hay had a significant (*p* < 0.05) effect on protein solubilization and microbial protein efficiency. Maximum (*p* < 0.05) solubilisation of protein was found when *A. pendula* (2:1) and *E. jambolana* (3:1) were supplemented with wheat straw (Table 3). It is noteworthy that the solubilization (%) of leaves (16.60 ± 1.97 of *A. pendula* and 15.03 ± 1.06 of *E. jambolana*) was comparatively higher (*p* < 0.05) than that of roughage wheat straw (8.95 ± 0.95) and berseem hay (3.04 ± 0.08) alone. 

The level of NRNAQ revealed that supplementation of *E. jambolana* and *A. pendula* leaves increased (*p* < 0.05) microbial protein synthesis in comparison to wheat straw and berseem hay (Table 3). Microbial synthesis of *E. jambolana* and *A. pendula* was higher (*p* < 0.05) than wheat straw and berseem hay alone. Supplementation of tree leaves to wheat straw or berseem hay at different ratios exhibited inconsistent effects on microbial protein synthesis, but supplementation increased (*p* < 0.05) overall microbial protein synthesis in contrast to wheat straw and berseem hay alone.

## 3. Discussion

In general, the activities of ruminal enzymes (*R*–GOT, *R*–GPT and *R*–cellulase) were inhibited due to the addition of PA-rich fractions during in vitro fermentation. The presence of (−) epigallocatechin and (+) catechin-4-phloroglucinol was not observed in AP–DW, which may be due to low solvent extraction efficiency. This study validated the antimicrobial nature of (+) catechin, (−) epicatechin and other proanthocyanidins derivatives isolated from *E. jambolana* and *A. pendula*. Similar effects of phenolic compounds on rumen microbes have been reported in other studies [27,28]. The other possible reason for the reduction of enzymatic activities might be the lower degrees of polymerization recorded (2–7) for the tree species chosen for the present study. The degree of polymerization has a positive correlation with antimicrobial activity, as described previously [29,30]. The role of the degree of polymerization can be explained by enhancing the stability of the metal–flavonoids complex, which is cytoprotective against superoxide radical scavenging [31,32]. In the case of cellulose digestion, the inhibitory effect of PAs has been reported [33]. Even though the mechanism of action of PAs is still unclear, they do modulate enzyme activity. The significant decrease in *R*–GDH activity from both the bacterial part and the protozoal part is consistent with earlier findings [34,35]. 

The significant decrease in the concentration of rumen protein upon addition of each extract with increasing incubation time and concentration was consistent with previous studies [36,37]. An explanation for these results may be steric interference at the binding of protease and the susceptible site during their interaction. The degree of binding also varies with types of PA and protein. Thus, PAs increase the quantity of protein flow from the rumen to the intestine, which then becomes available to the animal [38]. In legumes, PAs can be harmful if they lower the ruminal digestion of hemicellulose [39]. 

Solubilization and degradation are the two main processes which influence the degree of digestion of foliage in an animal gut. The degree of solubilization determines the release of protein from the plant cell to the ruminal fluid [40]. During the in vitro incubation, solubilization resembles rubisco loss from the feed substrate. It is suggested that the rate of degradation of rubisco by rumen microorganisms is influenced not only by the solubilization but also by the chemical structure of protein. The protein structure can interact with PAs and subsequently influence protein metabolism. The presence of PAs increases duodenal NAN, flow per unit of total N eaten, and reduces the ammonia concentration in the rumen [41]. Results from the current study show that the rubisco was solubilized more by supplementing the leaves of *E. jambolana* and *A. pendula* in different ratios. Similar results were observed by adding phenolics in *Lotus pedunculatus* [42] and in *Trifolium repens* [40]. The structure and molecular weight of PAs influence microbial protein synthesis. In the present study, supplementation of *E. jambolana* and *A. pendula* leaves to wheat straw and berseem hay increased microbial protein synthesis. Similar results were observed by supplementation of gallic acid, tannic acid or quebracho tannins to alfalfa hay [43]. In contrast, a reduction in microbial protein was found with maize supplementation [44], which could be due to the suppression of microbial growth.

## 4. Materials and Methods

### 4.1. Chemicals and Reagents

Catechin, epicatechin, gallocatechin, epigallocatechin, catechin-4-phloroglucinol, epicatechin-4-phloroglucinol, gallocatechin-4-phloroglucinol, epigallocatechin-4-phloroglucinol and Sephadex LH–20 of analytical grade were purchased from Sigma, USA. All other reagents and solvents used were of analytical grade. UV spectra were measured on a UNCAM UV/Vis spectrophotometer (Newington, CT, USA). TLC, column chromatography and paper chromatography were performed on precoated Si GF^256^, Si gel (60–120 Mesh, Merck, India).

### 4.2. Collection of Plant Material and Extraction of Proanthocyanidins (PAs)

Leaf samples from *E. jambolana* and *A. pendula* trees were collected from the Central Research Farm of ICAR-Indian Grassland and Fodder Research Institute and ICAR-Central Agro-Forestry Research Institute, Jhansi, India. Collected leaves were allowed to dry under the shade and then subjected to drying in hot air oven at 60 °C until the constant dry weight was reached. The dried leaves were powdered and sieved through 1 mm. About 2 kg of powder per species were soaked overnight in 70% acetone containing 0.1% ascorbic acid. After soaking, the solvent was removed under vacuum in a rotatory evaporator at 40 °C. The remaining aqueous phase was subsequently washed with pure chloroform, diethyl ether and ethyl acetate, and then the left-over extract was diluted with an equal amount (weight/volume) of methanol and charged over pre-equilibrated Sephadex LH–20 (30 × 2.5 cm). The gradient elution of the column with H_2_O and methanol (1:1) yielded compounds EJ–50 and AP–50 as a light-pink crystalline substance. Further, the column was eluted with acetone and H_2_O (7:3), yielding compounds EJ–70 and AP–70 as a light-coloured substance. Finally, the column was eluted with distilled H_2_O (DW), yielding compounds EJ–DW and AP–DW. A detailed view of the extraction procedure is drawn in a flowchart that is available in Appendix A. 

All extracted compounds were subjected to qualitative phytochemical investigation for the presence of phenolics and flavonoids through Shinoda, vanillin/HCL and FeCl_3_ tests along with Thin Layer Chromatography (TLC) and paper chromatography (PC) profiling [45]. To determine the monomeric unit, each isolated proanthocyanidin was independently treated with phloroglucinol in the presence of 1% HCl in ethyl alcohol for 4 h. After removal of solvent, the product was diluted with water and extracted with ethyl acetate followed by its evaporation to dryness. The dry product was dissolved in 80% methanol and subjected to quantitative analysis by HPTLC (cellulose), solvent system (tertiary butyl alcohol and 6% acetic acid) and HPLC (equipped with UV/VIS detector and RP ODS column (25 cm × 4 mm, id) at ambient temperature with a solvent system of acetic acid (1%) and methanol at 1 mL/min. The retention time (Rt) in HPLC ranged between 15.31 to 48.71 min for the above flavonoids. Quantification of flavonoids was accomplished by using regression equation Y = mχ + c, which was obtained from a calibration curve formed by using a standard solution of different dilutions of flavonoid mixtures. The limit of detection and limit of quantification were determined at signal-to-noise (S/N) ratios of 3 and 10, respectively. To study the subunits, each isolate was hydrolysed completely with 5% HCl in n-Butanol for 1.5 h in a boiling water bath. After removal of the acid, the resultant product was examined chromatographically on Whatman No. 1 and Whatman No. 3 and cellulose TLC plate using Forestal as the solvent system for their resolution. The resolved spots were identified by co-comparison of R_f_ value of pure samples obtained from Sigma Aldrich, USA. Estimates of anthocyanidin were determined by using UV spectroscopy.

### 4.3. Ruminal Microbial Enzyme Activity 

A male sheep species of local breed with mean body weight of 26.9 ± 2.1 kg was selected from a small ruminant unit of the Plant Animal Relationship Division of IGFRI, Jhansi to collect rumen liquor. The rumen liquor was collected before feeding through the mouth using a perforated plastic tube with light suction in a 0.5 L capacity pre-warmed thermos and did not cause any distress to the animals. To estimate the activity of extra-cellular enzymes (cellulases), a previously described method was used [46]. For intra-cellular enzymes (GPT: Glutamic pyruvic transaminase, GOT: Glutamic oxaloacetic transaminase and GDH: Glutamate dehydrogenase), another previously published method was used [47]. The collected sheep rumen liquor was centrifuged to fractionate bacteria and protozoa rich fractions. From these fractions, bacteria and protozoa rich enzyme extracts were prepared in 0.1 M phosphate buffer of pH 6.8 [48]. The activities of the GOT and GPT in both fractions were assayed according to a published method [49]. A previous method [50] was followed to measure the specific activity of the GDH enzyme. The oxidation of the glutamate in both microbial fractions (protozoal and bacterial) was assessed by the decrease in optical density by measuring the rate of change in absorbance at 340 nm caused by the reduction of di-phosphopyridine nucleotide (DPNH). The protein concentration of the enzyme extracts was estimated by the method of Lowry [51] to work out the specific activities of the assayed enzymes. The proteolytic activity of the bacteria was studied by estimating undigested protein from casein [52] with minor modification [53].

### 4.4. Rubisco Solubilization and Microbial Protein Synthesis

Lyophilized powder made of macerated leaves (500 mg) of *A. pendula* and *E. jambolana* in combination with wheat straw or berseem hay in ratios of 1:1, 2:1 and 3:1 was incubated at 37 °C with 40 mL of strained rumen liquor from grazing sheep and 10 mL of CO_2_ saturated buffer (mM 117NaHCO_3_, 26Na_2_HPO_4_, 8NaCl, 8KCl, 0.2CaCl_2_ and 0.3MgCl_2_) for 48 h in a full anaerobic environment, and filtered through a G-1 crucible. The filtrate was centrifuged at 500× *g* for 15 min to remove the feed particles and the supernatant was re-centrifuged at 20,000× *g* for 30 min at 4 °C. This process was repeated thrice. The residual pellet was washed with a large volume of 0.9% saline and preserved for determination of microbial protein and RNA after lyophilization. The residue collected in the crucible was used for Rubisco determination by the Bradford method [54] after protein extraction with protein buffer and SDS-PAGE electrophoresis. To determine the microbial protein efficiency, the prepared pellet from the filtrate was used for nitrogen determination by the Kjeldahl method and microbial purine as RNA [55]. The efficiency of the microbial protein was calculated as the microbial Nitrogen: Microbial RNA Quotient (NRNAQ).

### 4.5. Statistical Analysis

Microsoft Excel 2015 and R (Ri3864.0.3) were used for the statistical analysis. Analysis of variance (ANOVA) was performed in R to evaluate the measurements of enzymatic activities, and means are considered significantly different at *p* < 0.05 by post-Hoc analysis with Tukey’s test.

## 5. Conclusions

Leaf extracts from *E. jambolana* and *A. pendula* containing PAs were found to reduce or inhibit ruminal enzymatic activities. On the other hand, supplementation of tree leaves along with wheat straw and berseem hay in different ratios in the sheep diet increased the rubisco solubilization and enhanced the microbial protein synthesis. These results indicated a relationship between the chemical structure of proanthocyanidin and rumen enzymes’ activities, which may be employed to manipulate the rumen fermentation, predominantly the fiber (cellulose) and protein utilization (transaminases—GOT and GPT, deaminase—GDH). Therefore, PA-enriched leaf extracts could be promising promising feed supplement to manipulate ruminal enzyme activities for efficient and better utilisation of protein in ruminants. 

## Figures and Tables

**Figure 1 molecules-27-05870-f001:**
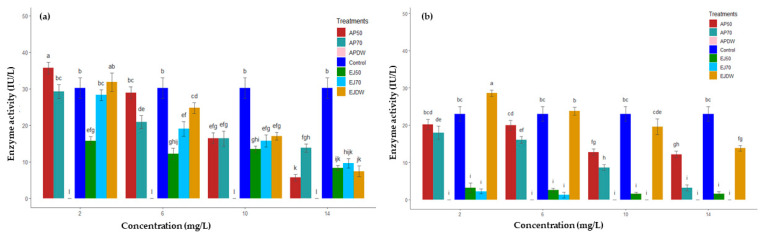
The effect of different proanthocyanin extracts AP–50, AP–70 and AP–DW: Extract eluted with 50% MeOH, 70% acetone and with distilled water, respectively, from *A. pendula*. EJ–50, EJ–70 and EJ–DW: Extract eluted with 50% of MeOH, 70% of acetone and with distilled water, respectively, from *E. jambolana* at different concentrations on ruminal glutamate dehydrogenase activity (*R*–GDH) (**a**) protozoal and (**b**) bacterial. Different letters associated with the bars indicate significant (*p* < 0.05) difference.

**Figure 2 molecules-27-05870-f002:**
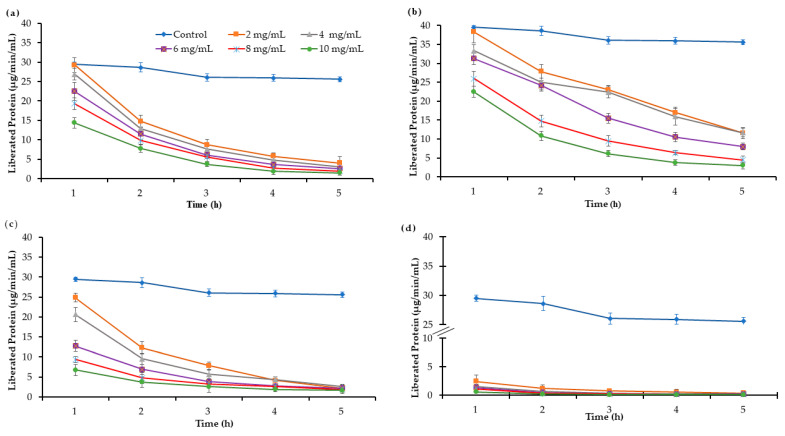
The effect of individual proanthocyanin extract on *R*–protease at different their concentrations and incubation times (**a**) AP–50, (**b**) AP–70, (**c**) EJ–70 and (**d**) EJ–50.

**Table 1 molecules-27-05870-t001:** Composition of oligomers of proanthocyanidins in tree leaves.

Components (mg/g)	AP–50	AP–70	AP–DW	EJ–50	EJ–70	EJ–DW
(+) Catechin	1.03 ± 0.07 ^a^	1.21 ± 0.38 ^a^	0.01 ± 0.00 ^c^	0.22 ± 0.01 ^bc^	0.12 ± 0.02 ^bc^	0.47 ± 0.04 ^b^
(−) Epicatechin	0.32 ± 0.04 ^ab^	0.44 ± 0.18 ^a^	0.44 ± 0.04 ^a^	0.12 ± 0.02 ^b^	0.24 ± 0.03 ^ab^	0.33 ± 0.01 ^ab^
(+) Gallocatechin	0.09 ± 0.01 ^b^	0.51 ± 0.05 ^a^	0.21 ± 0.02 ^b^	0.12 ± 0.08 ^b^	0.10 ± 0.08 ^b^	0.21 ± 0.03 ^b^
(−) Epigallocatechin	0.25 ± 0.05 ^ab^	0.23 ± 0.07 ^ab^	0.00 ± 0.00 ^c^	0.26 ± 0.06 ^ab^	0.14 ± 0.02 ^b^	0.35 ± 0.03 ^a^
(+) Catechin-4-phloroglucinol	3.58 ± 1.24 ^a^	4.40 ± 0.94 ^a^	0.00 ± 0.00 ^c^	3.30 ± 0.34 ^ab^	1.56 ± 0.10 ^bc^	3.96 ± 0.38 ^a^
(+) Gallocatechin-4-phloroglucinol	1.45 ± 0.09 ^b^	0.85 ± 0.13 ^c^	2.86 ± 0.12 ^a^	1.24 ± 0.11 ^b^	0.65 ± 0.09 ^c^	1.36 ± 0.09 ^b^
(−) Epigallocatechin-4-phloroglucinol	1.27 ± 0.10 ^b^	1.10 ± 0.12 ^bc^	1.12 ± 0.03 ^b^	0.86 ± 0.09 ^c^	1.02 ± 0.03 ^bc^	1.54 ± 0.13 ^a^

AP–50, AP–70 and AP–DW: Extract eluted with 50% of MeOH, 70% acetone and with distilled water respectively from *A. pendula*. EJ–50, EJ–70 and EJ–DW: Extract eluted with 50% of MeOH, 70% of acetone and with distilled water respectively from *E. jambolana.* Values are indicated as mean ± SD and values within the same row with different superscript letters are significantly different (*p* < 0.05).

**Table 2 molecules-27-05870-t002:** Effect of Proanthocyanidins from *A. pendula* and *E. jambolana* on inhibition activity (EC_50_) of ruminal glutamic oxaloacetic transaminase (*R*–GOT), ruminal glutamic pyruvic transaminase (*R*–GPT) and ruminal *R*–cellulase enzymes.

Enzymes EC_50_ (mg/mL)	Control	TA	GA	AP–50	AP–70	AP–DW	EJ–50	EJ–70	EJ–DW
*R*–GOT (P)	22.22 ± 0.64 ^a^	20.54 ± 1.13 ^ab^	21.77 ± 0.80 ^a^	20.27 ± 2.06 ^ab^	5.45 ± 0.71 ^d^	20.79 ± 1.03 ^ab^	18.5 ± 1.25 ^b^	13.78 ± 0.86 ^c^	4.44 ± 0.57 ^d^
*R*–GOT (B)	23.31 ± 2.09 ^b^	17.54 ± 0.13 ^cd^	22.78 ± 0.80 ^b^	9.02 ± 0.63 ^e^	19.58 ± 0.46 ^bc^	19.6 ± 1.78 ^bc^	34.24 ± 1.58 ^a^	18.39 ± 2.32 ^c^	14.05 ± 1.86 ^d^
*R*–GPT (P)	20.49 ± 1.21 ^a^	17.52 ± 0.70 ^d^	6.89 ± 0.12 ^e^	9.67 ± 1.32 ^e^	17.75 ± 1.92 ^cd^	16.23 ± 0.79 ^d^	24.03 ± 0.89 ^b^	20.66 ± 0.65 ^c^	7.83 ± 1.0 ^e^
*R*–GPT (B)	24.96 ± 1.74 ^a^	17.12 ± 0.15 ^c^	15.52 ± 0.53 ^c^	15.91 ± 0.24 ^c^	17.43 ± 0.71 ^c^	16.86 ± 1.43 ^c^	3.23 ± 0.26 ^d^	23 ± 1.03 ^ab^	21.04 ± 2.06 ^b^
*R*–Cellulase	82.84 ± 2.89 ^b^	124.32 ± 1.63 ^a^	6.62 ± 0.56 ^d^	9.08 ± 1.02 ^d^	23.36 ± 1.34 ^c^	81.09 ± 1.54 ^b^	21.21 ± 1.86 ^c^	82.50 ± 2.32 ^b^	–

*R*–GOT: Ruminal glutamic oxaloacetic transaminase, *R*–GPT: Ruminal pyruvic transaminase, *R*–cellulase: Ruminal cellulase, P: Protozoal, B: Bacterial, TA: Tannic acid, GA: Gallic acid. AP–50, AP–70 and AP–DW: Extract eluted with 50% of MeOH, 70% acetone and with distilled water respectively from *A. pendula*. EJ–50, EJ–70 and EJ–DW: Extract eluted with 50% of MeOH, 70% of acetone and with distilled water respectively from *E. jambolana.* Values are indicated as mean ± SD and values within the same row with different superscript letters are significantly different (*p* < 0.05).

**Table 3 molecules-27-05870-t003:** Level of microbial nitrogen and RNA quotient (NRNAQ) and % Solubilization of Rubisco in ruminal substrate after in vitro digestion of tree leaves with wheat straw (WS) and berseem hay (BH).

Diet		Solubilization (%)	NRNAQ
AP		16.60 ± 1.97 ^c^	0.25 ± 0.01 ^fgh^
EJ		15.03 ± 1.06 ^cd^	0.26 ± 0.02 ^efg^
WS		8.95 ± 0.95 ^e^	0.16 ± 0.01 ^i^
BH		3.04 ± 0.08 ^f^	0.17 ± 0.01 ^i^
AP + WS	1:1	15.39 ± 1.67 ^c^	0.42 ± 0.03 ^a^
	2:1	31.46 ± 1.06 ^a^	0.33 ± 0.01 ^cd^
	3:1	23.05 ± 0.51 ^b^	0.16 ± 0.01 ^i^
AP + BH	1:1	10.55 ± 0.83 ^e^	0.37 ± 0.01 ^bc^
	2:1	11.30 ± 0.28 ^e^	0.41 ± 0.03 ^ab^
	3:1	8.87 ± 1.21 ^e^	0.31 ± 0.02 ^de^
EJ + WS	1:1	10.77 ± 1.45 ^e^	0.25 ± 0.01 ^fgh^
	2:1	23.28 ± 1.23 ^b^	0.23 ± 0.01 ^gh^
	3:1	28.19 ± 0.62 ^a^	0.20 ± 0.01 ^hi^
EJ + BH	1:1	11.75 ± 1.55 ^de^	0.32 ± 0.03 ^cd^
	2:1	9.76 ± 0.87 ^e^	0.29 ± 0.01 ^def^
	3:1	9.71 ± 0.76 ^e^	0.24 ± 0.02 ^fgh^

AP: *A. pendula*, EJ: *E. jambolana*, WS: Wheat straw, BH: Berseem hay, NRNAQ: Nitrogen-RNA Quotient. Values are indicated as mean ± SD and values within the same column with different superscript letters are significantly different (*p* < 0.05).

## Data Availability

Not applicable.

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
