# Peer review of "Proanthocyanidins Modulate Rumen Enzyme Activities and Protein Utilization In Vitro"

_molecules, 2022, doi:10.3390/molecules27185870_

Round 1

Reviewer 1 Report

In one of the authors, Manjree Agarwal, 2 and 3 are super scripted, but in the address, there is no 3. Kindly check and correct.

The comments are attached

Reviewer 2 Report

Proanthocyanidins were extracted from leaves of A. pendula (AP) and E. jambolana (EJ), and six fractions of proanthocyanidins were obtained. Then, the obtained fractions were examined for their molecular structures and heir effects on sheep ruminal enzymes and the solubilization of principal leaf protein rubisco in vitro. Furthermore, the effects of these fractions on rubisco activity were also studied in relation to their molecular structures. Consequently, the obtained fractions were shown to inhibit various ruminal enzymes, glutamic oxaloacetic transaminase, glutamic pyruvic transaminase, and several fractions were also shown to significantly inhibit the activity of R-cellulase. Most of the fractions inhibited R-glutamate dehydrogenase activity by increasing their concentrations. By contrast, the fractions decreased protease activity up to 58% with increasing incubation time and their concentrations. Thus, the obtained results were  considered to provide evidence for suggesting that proanthocyanidins obtained from E.J. and A.P. leaves exhibited inhibitory activities on rumen enzymes and higher protein solubilization, thereby having the potentials to modulate rumen enzymes activities for efficient utilization of protein and fiber in ruminants.

The subject of this study, particularly the objects handling in this research seemed unique, and possibly considered to be significant only in specialized scientific field. However, it would be quite practical and meaningful for specific people in particular region. This manuscript was considered to report simply the results of analyses and measurements, and therefore, there was no logical defects observed. The experiments were designed properly, and carried out without any particular and critical defects. The results were presented without any a hitch, and the interpretation of obtained results was reasonable and appropriate.

Manuscript seemed to be written without any major serious problems, but it would probably be necessary to rethink usage of prepositions and idioms. In addition, the wording and the  expression should be refined a little more. For example, the phrase “in combination to” in lines 51-52 should be “in combination with”. Also, the phrase “The results of our findings of” in line 72 was awkward, and it should be “The results of our studies on (The results of our analyses of). On the whole, the manuscript had no serious problem, but English could be considered to refine a little more.

Anyway, there were small mistakes and flaws probably due to carelessness, and it would be strongly recommend to read the manuscript with utmost care. So it would be recommended to make minor revision.

Author Response

Dear Reviewer,

Thanks for your valuable suggestions and comments to improve our manuscript.

  1. I have included your suggestions and refined the sentences in line number 51-52 and line number 72.
  2. In addition to this, I have re-read the entire manuscript and revised minor mistakes. 

Kind regards